# Optimistic Bandit Convex Optimization

**Mehryar Mohri**
Courant Institute and Google
251 Mercer Street
New York, NY 10012
`mohri@cims.nyu.edu`

**Scott Yang**
Courant Institute
251 Mercer Street
New York, NY 10012
`yangs@cims.nyu.edu`

## Abstract

We introduce the general and powerful scheme of predicting information re-use in optimization algorithms. This allows us to devise a computationally efficient algorithm for bandit convex optimization with new state-of-the-art guarantees for both Lipschitz loss functions and loss functions with Lipschitz gradients. This is the first algorithm admitting both a polynomial time complexity and a regret that is polynomial in the dimension of the action space that improves upon the original regret bound for Lipschitz loss functions, achieving a regret of $\widetilde{\mathcal{O}}\left(T^{11/16}d^{3/8}\right)$. Our algorithm further improves upon the best existing polynomial-in-dimension bound (both computationally and in terms of regret) for loss functions with Lipschitz gradients, achieving a regret of $\widetilde{\mathcal{O}}\left(T^{8/13}d^{5/3}\right)$.

## 1 Introduction

Bandit convex optimization (BCO) is a key framework for modeling learning problems with sequential data under partial feedback. In the BCO scenario, at each round, the learner selects a point (or action) in a bounded convex set and observes the value at that point of a convex loss function determined by an adversary. The feedback received is limited to that information: no gradient or any other higher order information about the function is provided to the learner. The learner's objective is to minimize his regret, that is the difference between his cumulative loss over a finite number of rounds and that of the loss of the best fixed action in hindsight.

The limited feedback makes the BCO setup relevant to a number of applications, including online advertising. On the other hand, it also makes the problem notoriously difficult and requires the learner to find a careful trade-off between exploration and exploitation. While it has been the subject of extensive study in recent years, the fundamental BCO problem remains one of the most challenging scenarios in machine learning where several questions concerning optimality guarantees remain open.

The original work of Flaxman et al. [2005] showed that a regret of $\widetilde{\mathcal{O}}(T^{5/6})$ is achievable for bounded loss functions and of $\widetilde{\mathcal{O}}(T^{3/4})$ for Lipschitz loss functions (the latter bound is also given in [Kleinberg, 2004]), both of which are still the best known results given by explicit algorithms. Agarwal et al. [2010] introduced an algorithm that maintains a regret of $\widetilde{\mathcal{O}}(T^{2/3})$ for loss functions that are both Lipschitz and strongly convex, which is also still state-of-the-art. For functions that are Lipschitz and also admit Lipschitz gradients, Saha and Tewari [2011] designed an algorithm with a regret of $\widetilde{\mathcal{O}}(T^{2/3})$ regret, a result that was recently improved to $\widetilde{\mathcal{O}}(T^{5/8})$ by Dekel et al. [2015].

Here, we further improve upon these bounds both in the Lipschitz and Lipschitz gradient settings. By incorporating the novel and powerful idea of predicting information re-use, we introduce an algorithm with a regret bound of $\widetilde{\mathcal{O}}\left(T^{11/16}\right)$ for Lipschitz loss functions. Similarly, our algorithm also achieves the best regret guarantee among computationally tractable algorithms for loss functions with Lipschitz

gradients: $\widetilde{\mathcal{O}}\left(T^{8/13}\right)$. Both bounds admit a relatively mild dependency on the dimension of the action space.

We note that the recent remarkable work by [Bubeck et al., 2015, Bubeck and Eldan, 2015] has proven the existence of algorithms that can attain a regret of $\widetilde{\mathcal{O}}(T^{1/2})$, which matches the known lower bound $\Omega(T^{1/2})$ given by Dani et al.. Thus, the dependency of our bounds with respect to $T$ is not optimal. Furthermore, two recent unpublished manuscripts, [Hazan and Li, 2016] and [Bubeck et al., 2016], present algorithms achieving regret $\widetilde{\mathcal{O}}(T^{1/2})$. These results, once verified, would be ground-breaking contributions to the literature. However, unlike our algorithms, the regret bound for both of these algorithms admits a large dependency on the dimension $d$ of the action space: exponential for [Hazan and Li, 2016], $d^{\mathcal{O}(9.5)}$ for [Bubeck et al., 2016]. One hope is that the novel ideas introduced by Hazan and Li [2016] (the application of the ellipsoid method with a *restart button* and lower convex envelopes) or those by Bubeck et al. [2016] (which also make use of the restart idea but introduces a very original kernel method) could be combined with those presented in this paper to derive algorithms with the most favorable guarantees with respect to both $T$ and $d$.

We begin by formally introducing our notation and setup. We then highlight some of the essential ideas in previous work before introducing our new key insight. Next, we give a detailed description of our algorithm for which we prove theoretical guarantees in several settings.

## 2    Preliminaries

### 2.1    BCO scenario

The scenario of bandit convex optimization, which dates back to [Flaxman et al., 2005], is a sequential prediction problem on a convex compact domain $\mathcal{K} \subset \mathbb{R}^d$. At each round $t \in [1, T]$, the learner selects a (possibly) randomized action $x_t \in \mathcal{K}$ and incurs the loss $f_t(x_t)$ based on a convex function $f_t \colon \mathcal{K} \to \mathbb{R}$ chosen by the adversary. We assume that the adversary is oblivious, so that the loss functions are independent of the player's actions. The objective of the learner is to minimize his regret with respect to the optimal static action in hindsight, that is, if we denote by $\mathcal{A}$ the learner's randomized algorithm, the following quantity:

$$\mathrm{Reg}_T(\mathcal{A}) = \mathbb{E}\left[\sum_{t=1}^{T} f_t(x_t)\right] - \min_{x \in \mathcal{K}} \sum_{t=1}^{T} f_t(x). \tag{1}$$

We will denote by $D$ the diameter of the action space $\mathcal{K}$ in the Euclidean norm: $D = \sup_{x,y \in \mathcal{K}} \|x - y\|_2$. Throughout this paper, we will often use different induced norms. We will denote by $\| \cdot \|_A$ the norm induced by a symmetric positive definite (SPD) matrix $A \succ 0$, defined for all $x \in \mathbb{R}^d$ by $\|x\|_A = \sqrt{x^\top A x}$. Moreover, we will denote by $\| \cdot \|_{A,*}$ its dual norm, given by $\| \cdot \|_{A^{-1}}$. To simplify the notation, we will write $\| \cdot \|_x$ instead of $\| \cdot \|_{\nabla^2 \mathcal{R}(x)}$, when the convex and twice differentiable function $\mathcal{R} \colon \mathrm{int}(\mathcal{K}) \to \mathbb{R}$ is clear from the context. Here, $\mathrm{int}(\mathcal{K})$ is the set interior of $\mathcal{K}$.

We will consider different levels of regularity for the functions $f_t$ selected by the adversary. We will always assume that they are uniformly bounded by some constant $C > 0$, that is $|f_t(x)| \leq C$ for all $t \in [1, T]$ and $x \in \mathcal{K}$, and, by shifting the loss functions upwards by at most $C$, we will also assume, without loss of generality, that they are non-negative: $f_t \geq 0$, for all $t \in [1, T]$. Moreover, we will always assume that $f_t$ is Lipschitz on $\mathcal{K}$ (henceforth denoted $\mathcal{C}^{0,1}(\mathcal{K})$):

$$\forall t \in [1, T], \, \forall x, y \in \mathcal{K}, \quad |f_t(x) - f_t(y)| \leq L\|x - y\|_2.$$

In some instances, we will further assume that the functions admit $H$-Lipschitz gradients on the interior of the domain (henceforth denoted $\mathcal{C}^{1,1}(\mathrm{int}(\mathcal{K}))$):

$$\exists H > 0 \colon \forall t \in [1, T], \, \forall x, y \in \mathrm{int}(\mathcal{K}), \quad \|\nabla f_t(x) - \nabla f_t(y)\|_2 \leq H\|x - y\|_2.$$

Since $f_t$ is convex, it admits a subgradient at any point in $\mathcal{K}$. We denote by $g_t$ one element of the subgradient at the point $x_t \in \mathcal{K}$ selected by the learner at round $t$. When the losses are $\mathcal{C}^{1,1}$, the only element of the subgradient is the gradient, and $g_t = \nabla f_t(x_t)$. We will use the shorthand $v_{1:t} = \sum_{s=1}^{t} v_s$ to denote the sum of $t$ vectors $v_1, \ldots, v_t$. In particular, $g_{1:t}$ will denote the sum of the subgradients $g_s$ for $s \in [1, t]$.

Lastly, we will denote by $B_1(0) = \left\{x \in \mathbb{R}^d \colon \|x\|_2 \leq 1\right\} \subset \mathbb{R}^d$ the $d$-dimensional Euclidean ball of radius one and by $\partial B_1(0)$ the unit sphere.

## 2.2 Follow-the-regularized-leader template

A standard algorithm in online learning, both for the bandit and full-information setting is the follow-the-regularized-leader (FTRL) algorithm. At each round, the algorithm selects the action that minimizes the cumulative linearized loss augmented with a regularization term $R\colon \mathcal{K} \to \mathbb{R}$. Thus, the action $x_{t+1}$ is defined as follows:

$$x_{t+1} = \operatorname*{argmin}_{x \in \mathcal{K}} \eta g_{1:t}^\top x + R(x),$$

where $\eta > 0$ is a learning rate that determines the tradeoff between greedy optimization and regularization.

If we had access to the subgradients at each round, then, FTRL with $R(x) = \|x\|_2^2$ and $\eta = \frac{1}{\sqrt{T}}$ would yield a regret of $\mathcal{O}(\sqrt{dT})$, which is known to be optimal. But, since we only have access to the loss function values $f_t(x_t)$ and since the loss functions change at each round, a more refined strategy is needed.

### 2.2.1 One-point gradient estimates and surrogate losses

One key insight into the bandit convex optimization problem, due to Flaxman et al. [2005], is that the subgradient of a smoothed version of the loss function can be estimated by sampling and rescaling around the point the algorithm originally intended to play.

**Lemma 1** ([Flaxman et al., 2005, Saha and Tewari, 2011]). *Let $f\colon \mathcal{K} \to \mathbb{R}$ be an arbitrary function (not necessarily differentiable) and let $U(\partial B_1(0))$ denote the uniform distribution over the unit sphere. Then, for any $\delta > 0$ and any SPD matrix $A \succ 0$, the function $\widehat{f}$ defined for all $x \in \mathcal{K}$ by $\widehat{f}(x) = \mathbb{E}_{u \sim U(\partial B_1(0))}[f(x + \delta Au)]$ is differentiable over $\operatorname{int}(\mathcal{K})$ and, for any $x \in \operatorname{int}(\mathcal{K})$, $\widehat{g} = \frac{d}{\delta} f(x + \delta Au) A^{-1} u$ is an unbiased estimate of $\nabla \widehat{f}(x)$:*

$$\mathbb{E}_{u \sim U(\partial B_1(0))} \left[ \frac{d}{\delta} f(x + \delta Au) A^{-1} u \right] = \nabla \widehat{f}(x).$$

The result shows that if at each round $t$ we sample $u_t \sim U(\partial B_1(0))$, define an SPD matrix $A_t$ and play the point $y_t = x_t + \delta A_t u$ (assuming that $y_t \in \mathcal{K}$), then $\widehat{g}_t = \frac{d}{\delta} f(x_t + \delta A_t u_t) A_t^{-1} u_t$ is an unbiased estimate of the gradient of $\widehat{f}$ at the point $x_t$ originally intended: $\mathbb{E}[\widehat{g}_t] = \nabla \widehat{f}(x_t)$. Thus, we can use FTRL with these smoothed gradient estimates: $x_{t+1} = \operatorname{argmin}_{x \in \mathcal{K}} \eta \widehat{g}_{1:t}^\top x + R(x)$, at the cost of the approximation error from $f_t$ to $\widehat{f}_t$. Furthermore, the norm of these estimate gradients can be bounded.

**Lemma 2.** *Let $\delta > 0$, $u_t \in \partial B_1(0)$ and $A_t \succ 0$, then the norm of $\widehat{g}_t = \frac{d}{\delta} f(x_t + \delta A_t u_t) A_t^{-1} u_t$ can be bounded as follows: $\|\widehat{g}_t\|_{A_t^2}^2 \leq \frac{d^2}{\delta^2} C^2$.*

*Proof.* Since $f_t$ is bounded by $C$, we can write $\|\widehat{g}_t\|_{A_t^2}^2 \leq \frac{d^2}{\delta^2} C^2 u_t A_t^{-1} A_t^2 A_t^{-1} u_t \leq \frac{d^2}{\delta^2} C^2$. $\qquad\square$

This gives us a bound on the Lipschitz constant of $\widehat{f}_t$ in terms of $d$, $\delta$, and $C$.

### 2.2.2 Self-concordant barrier as regularization

When sampling to derive a gradient estimate, we need to ensure that the point sampled lies within the feasible set $\mathcal{K}$. A second key idea in the BCO problem, due to Abernethy et al. [2008], is to design ellipsoids that are always contained in the feasible sets. This is done by using tools from the theory of interior-point methods in convex optimization.

**Definition 1** (Definition 2.3.1 [Nesterov and Nemirovskii, 1994]). *Let $\mathcal{K} \subset \mathbb{R}^d$ be closed convex, and let $\nu \geq 0$. A $C^3$ function $\mathcal{R}\colon \operatorname{int}(\mathcal{K}) \to \mathbb{R}$ is a $\nu$-self-concordant barrier for $\mathcal{K}$ if for any sequence $(z_s)_{s=1}^\infty$ with $z_s \to \partial \mathcal{K}$, we have $\mathcal{R}(z_s) \to \infty$, and if for all $x \in \operatorname{int}(\mathcal{K})$, and $y \in \mathbb{R}^d$, the following inequalities hold:*

$$|\nabla^3 \mathcal{R}(x)[y, y, y]| \leq 2\|y\|_x^3, \quad |\nabla \mathcal{R}(x)^\top y| \leq \nu^{1/2} \|y\|_x.$$

Since self-concordant barriers are preserved under translation, we will always assume for convenience that $\min_{x \in \mathcal{K}} \mathcal{R}(x) = 0$.

Nesterov and Nemirovskii [1994] show that any $d$-dimensional closed convex set admits an $\mathcal{O}(d)$-self-concordant barrier. This allows us to always choose a self-concordant barrier as regularization.

We will use several other key properties of self-concordant barriers in this work, all of which are stated precisely in Appendix 7.1.

## 3 Previous work

The original paper by Flaxman et al. [2005] sampled indiscriminately around spheres and projected back onto the feasible set at each round. This yielded a regret of $\widetilde{\mathcal{O}}\left(T^{3/4}\right)$ for $\mathcal{C}^{0,1}$ loss functions. The follow-up work of Saha and Tewari [2011] showed that for $\mathcal{C}^{1,1}$ loss functions, one can run FTRL with a self-concordant barrier as regularization and sample around the Dikin ellipsoid to attain an improved regret bound of $\widetilde{\mathcal{O}}\left(T^{2/3}\right)$.

More recently, Dekel et al. [2015] showed that by averaging the smoothed gradient estimates and still using the self-concordant barrier as regularization, one can achieve a regret of $\widetilde{\mathcal{O}}\left(T^{5/8}\right)$. Specifically, denote by $\bar{g}_t = \frac{1}{k+1} \sum_{i=0}^{k} \widehat{g}_{t-i}$ the average of the past $k+1$ incurred gradients, where $\widehat{g}_{t-i} = 0$ for $t - i \leq 0$. Then we can play FTRL on these averaged smoothed gradient estimates: $x_{t+1} = \operatorname{argmin}_{\in \mathcal{K}} \eta \bar{g}_t^\top x + R(x)$, to attain the better guarantee.

Abernethy and Rakhlin [2009] derive a generic estimate for FTRL algorithms with self-concordant barriers as regularization:

**Lemma 3** ([Abernethy and Rakhlin, 2009]-Theorem 2.2-2.3). *Let $\mathcal{K}$ be a closed convex set in $\mathbb{R}^d$ and let $\mathcal{R}$ be a $\nu$-self-concordant barrier for $\mathcal{K}$. Let $\{g_t\}_{t=1}^{T} \subset \mathbb{R}^d$ and $\eta > 0$ be such that $\eta\|g_t\|_{x_t,*} \leq 1/4$ for all $t \in [1, T]$. Then, the FTRL update $x_{t+1} = \operatorname{argmin}_{x \in \mathcal{K}} g_{1:t}^\top x + \mathcal{R}(x)$ admits the following guarantees:*

$$\|x_t - x_{t+1}\|_{x_t} \leq 2\eta\|g_t\|_{x_t,*}, \qquad \forall x \in \mathcal{K}, \ \sum_{t=1}^{T} g_t^\top (x_t - x) \leq 2\eta \sum_{t=1}^{T} \|g_t\|_{x_t,*}^2 + \frac{1}{\eta}\mathcal{R}(x).$$

By Lemma 2, if we use FTRL with smoothed gradients, then the upper bound in this lemma can be further bounded by

$$2\eta \sum_{t=1}^{T} \|\widehat{g}_t\|_{x_t,*}^2 + \frac{1}{\eta}\mathcal{R}(x) \leq 2\eta T \frac{C^2 d^2}{\delta^2} + \frac{1}{\eta}\mathcal{R}(x).$$

Furthermore, the regret is then bounded by the sum of this upper bound and the cost of approximating $f_t$ with $\widehat{f}_t$. On the other hand, Dekel et al. [2015] showed that if we used FTRL with averaged smoothed gradients instead, then the upper bound in this lemma can be bounded as

$$2\eta \sum_{t=1}^{T} \|\bar{g}_t\|_{x_t,*}^2 + \frac{1}{\eta}\mathcal{R}(x) \leq 2\eta T \left( \frac{32 C^2 d^2}{\delta^2 (k+1)} + 2 D^2 L^2 \right) + \frac{1}{\eta}\mathcal{R}(x).$$

The extra factor $(k+1)$ in the denominator, at the cost of now approximating $f_t$ with $\bar{f}_t$, is what contributes to their improved regret result.

In general, finding surrogate losses that can both be approximated accurately and admit only a mild variance is a delicate task, and it is not clear how the constructions presented above can be improved.

## 4 Algorithm

### 4.1 Predicting the predictable

Rather than designing a newer and better surrogate loss, our strategy will be to exploit the structure of the current state-of-the-art method. Specifically, we draw upon the technique of predictable sequences from [Rakhlin and Sridharan, 2013]. The idea here is to allow the learner to preemptively "guess" the

gradient at the next step and optimize for this in the FTRL update. Specifically, if $\widetilde{g}_{t+1}$ is an estimate of the time $t+1$ gradient $g_{t+1}$ based on information up to time $t$, then the learner should play:

$$x_{t+1} = \operatorname*{argmin}_{x \in \mathcal{K}} (g_{1:t} + \widetilde{g}_{t+1})^\top x + \mathcal{R}(x).$$

This *optimistic FTRL* algorithm admits the following guarantee:

**Lemma 4** (Lemma 1 [Rakhlin and Sridharan, 2013])**.** *Let $\mathcal{K}$ be a closed convex set in $\mathbb{R}^d$, and let $\mathcal{R}$ be a $\nu$-self-concordant barrier for $\mathcal{K}$. Let $\{g_t\}_{t=1}^T \subset \mathbb{R}^d$ and $\eta > 0$ such that $\eta \|g_t - \widetilde{g}_t\|_{x_t,*} \leq 1/4$ $\forall t \in [1, T]$. Then the FTRL update $x_{t+1} = \operatorname{argmin}_{x \in \mathcal{K}} (g_{1:t} + \widetilde{g}_{t+1})^\top x + \mathcal{R}(x)$ admits the following guarantee:*

$$\forall x \in \mathcal{K}, \ \sum_{t=1}^T g_t^\top (x_t - x) \leq 2\eta \sum_{t=1}^T \|g_t - \widetilde{g}_t\|_{x_t,*}^2 + \frac{1}{\eta} \mathcal{R}(x).$$

In general, it is not clear what would be a good prediction candidate. Indeed, this is why Rakhlin and Sridharan [2013] called this algorithm an "optimistic" FTRL. However, notice that if we elect to play the averaged smoothed losses as in [Dekel et al., 2015], then the update at each time is $\bar{g}_t = \frac{1}{k+1} \sum_{i=0}^k \widehat{g}_{t-i}$. This implies that the time $t+1$ gradient is $\bar{g}_{t+1} = \frac{1}{k+1} \sum_{i=0}^k \widehat{g}_{t+1-i}$, which includes the smoothed gradients from time $t+1$ down to time $t - (k-1)$. The key insight here is that at time $t$, all but the $(t+1)$-th gradient are known!

This means that if we predict

$$\widetilde{g}_{t+1} = \frac{1}{k+1} \sum_{i=0}^k \widehat{g}_{t+1-i} - \frac{1}{k+1} \widehat{g}_{t+1} = \frac{1}{k+1} \sum_{i=1}^k \widehat{g}_{t+1-i},$$

then the first term in the bound of Lemma 4 will be in terms of

$$g_t - \widetilde{g}_t = \frac{1}{k+1} \sum_{i=0}^k \widehat{g}_{t-i} - \frac{1}{k+1} \sum_{i=1}^k \widehat{g}_{t-i} = \frac{1}{k+1} \widehat{g}_t.$$

In other words, all but the time $t$ smoothed gradient will cancel out. Essentially, we are predicting the predictable portion of the averaged gradient and guaranteeing that the optimism will pay off. Moreover, where we gained a factor of $\frac{1}{k+1}$ in the averaged loss case, we should expect to gain a factor of $\frac{1}{(k+1)^2}$ by using this optimistic prediction.

Note that this technique of optimistically predicting the variance reduction is widely applicable. As alluded to with the reference to [Schmidt et al., 2013], many variance reduction-type techniques, particularly in stochastic optimization, use historical information in their estimates (e.g. SVRG [Johnson and Zhang, 2013], SAGA [Defazio et al., 2014]). In these cases, it is possible to "predict" the information re-use and improve the convergence rates of each algorithm.

## 4.2 Description and pseudocode

Here, we give a detailed description of our algorithm, OPTIMISTICBCO. At each round $t$, the algorithm uses a sample $u_t$ from the uniform distribution over the unit sphere to define an unbiased estimate of the gradient of $\widehat{f}_t$, a smoothed version of the loss function $f_t$, as described in Section 2.2.1: $\widehat{g}_t \leftarrow \frac{d}{\delta} f_t(y_t)(\nabla^2 \mathcal{R}(x_t))^{-1/2} u_t$. Next, the trailing average of these unbiased estimates over a fixed window of length $k+1$ is computed: $\bar{g}_t = \frac{1}{k+1} \sum_{i=0}^k \widehat{g}_{t-i}$. The remaining steps executed at each round coincide with the Follow-the-Regularized-Leader update with a self-concordant barrier used as a regularizer, augmented with an optimistic prediction of the next round's trailing average. As described in Section 4.1, all but one of the terms in the trailing average are known and we *predict* their occurence:

$$\widetilde{g}_{t+1} = \frac{1}{k+1} \sum_{i=1}^k \widehat{g}_{t+1-i}, \quad x_{t+1} = \operatorname*{argmin}_{x \in \mathcal{K}} \eta \left( \bar{g}_{1:t} + \widetilde{g}_{t+1} \right)^\top x + \mathcal{R}(x).$$

Note that Theorem 3 implies that the actual point we play, $y_t$, is always a feasible point in $\mathcal{K}$. Figure 1 presents the pseudocode of the algorithm.

$\text{OPTIMISTICBCO}(\mathcal{R}, \delta, \eta, k, x_1)$

```
1   for t ← 1 to T do
2       u_t ← SAMPLE(U(∂B_1(0)))
3       y_t ← x_t + δ(∇²R(x_t))^{-½} u_t
4       PLAY(y_t)
5       f_t(y_t) ← RECEIVELOSS(y_t)
6       ĝ_t ← (d/δ) f_t(y_t)(∇²R(x_t))^{-½} u_t
7       ḡ_t ← (1/(k+1)) Σ_{i=0}^{k} ĝ_{t-i}
8       g̃_{t+1} ← (1/(k+1)) Σ_{i=1}^{k} ĝ_{t+1-i}
9       x_{t+1} ← argmin_{x∈K} η(ḡ_{1:t} + g̃_{t+1})^⊤ x + R(x)
10  return Σ_{t=1}^{T} f_t(y_t)
```

Figure 1: Pseudocode of OPTIMISTICBCO, with $\mathcal{R}\colon \mathrm{int}(\mathcal{K}) \to \mathbb{R}$, $\delta \in (0,1]$, $\eta > 0$, $k \in \mathbb{Z}$, and $x_1 \in \mathcal{K}$.

## 5  Regret guarantees

In this section, we state our main results, which are regret guarantees for OPTIMISTICBCO in the $\mathcal{C}^{0,1}$ and $\mathcal{C}^{1,1}$ cases. We also highlight the analysis and proofs for each regime.

### 5.1  Main results

The following is our main result for the $\mathcal{C}^{0,1}$ case.

**Theorem 1** ($\mathcal{C}^{0,1}$ Regret)**.** *Let $\mathcal{K} \subset \mathbb{R}^d$ be a convex set with diameter $D$ and $(f_t)_{t=1}^{T}$ a sequence of loss functions with each $f_t\colon \mathcal{K} \to \mathbb{R}_+$ $C$-bounded and $L$-Lipschitz. Let $\mathcal{R}$ be a $\nu$-self-concordant barrier for $\mathcal{K}$. Then, for $\eta k \leq \frac{\delta}{12Cd}$, the regret of OPTIMISTICBCO can be bounded as follows:*

$$\mathrm{Reg}_T(\text{OPTIMISTICBCO}) \leq \epsilon L T + L\delta D T + \frac{Ck}{2} + \frac{2Cd^2\eta T}{\delta^2(k+1)^2} + \frac{1}{\eta}\log(1/\epsilon)$$
$$+ LT 2\eta D \left[ \sqrt{3}L^{1/2} + \sqrt{2}DLk + \frac{\sqrt{48}d\sqrt{k}}{\delta} \right].$$

*In particular, for $\eta = T^{-11/16}d^{-3/8}$, $\delta = T^{-5/16}d^{3/8}$, $k = T^{1/8}d^{1/4}$, the following guarantee holds for the regret of the algorithm:*

$$\mathrm{Reg}_T(\text{OPTIMISTICBCO}) = \widetilde{\mathcal{O}}\left( T^{11/16}d^{3/8} \right).$$

The above result is the first improvement on the regret of Lipschitz losses in terms of $T$ since the original algorithm of Flaxman et al. [2005] that is realizable from a concrete algorithm as well as polynomial in both dimension and time (both computationally and in terms of regret).

**Theorem 2** ($\mathcal{C}^{1,1}$ Bound)**.** *Let $\mathcal{K} \subset \mathbb{R}^d$ be a convex set with diameter $D$ and $(f_t)_{t=1}^{T}$ a sequence of loss functions with each $f_t\colon \mathcal{K} \to \mathbb{R}_+$ $C$-bounded, $L$-Lipschitz and $H$-smooth. Let $\mathcal{R}$ be a $\nu$-self-concordant barrier for $\mathcal{K}$. Then, for $\eta k \leq \frac{\delta}{12d}$, the regret of OPTIMISTICBCO can be bounded as follows:*

$$\mathrm{Reg}_T(\text{OPTIMISTICBCO}) \leq \epsilon L T + H\delta^2 D^2 T$$
$$+ (TL + DHT)2\eta k D \left[ \frac{\sqrt{3}L^{1/2}}{k} + \sqrt{2}DL + \frac{\sqrt{48}d}{\sqrt{k}\delta} \right] + \frac{1}{\eta}\log(1/\epsilon) + Ck + \eta\frac{d^2 T}{\delta^2(k+1)^2}.$$

*In particular, for $\eta = T^{-8/13}d^{-5/6}$, $\delta = T^{-5/26}d^{1/3}$, $k = T^{1/13}d^{5/3}$, the following guarantee holds for the regret of the algorithm:*

$$\mathrm{Reg}_T(\text{OPTIMISTICBCO}) = \widetilde{\mathcal{O}}\left( T^{8/13}d^{5/3} \right).$$

This result is currently the best polynomial-in-time regret bound that is also polynomial in the dimension of the action space (both computationally and in terms of regret). It improves upon the work of Saha and Tewari [2011] and Dekel et al. [2015].

We now explain the analysis of both results, starting with Theorem 1 for $\mathcal{C}^{0,1}$ losses.

## 5.2 $\mathcal{C}^{0,1}$ analysis

Our analysis proceeds in two steps. We first modularize the cost of approximating the original losses $f_t(y_t)$ incurred with the averaged smoothed losses that we treat as surrogate losses. Then we show that the algorithm minimizes the regret against the surrogate losses effectively. The proofs of all lemmas in this section are presented in Appendix 7.2.

**Lemma 5** ($\mathcal{C}^{0,1}$ Structural bound on true losses in terms of smoothed losses). *Let $(f_t)_{t=1}^T$ be a sequence of loss functions, and assume that $f_t \colon \mathcal{K} \to \mathbb{R}_+$ is $C$-bounded and $L$-Lipschitz, where $\mathcal{K} \subset \mathbb{R}^d$. Denote*

$$\widehat{f}_t(x) = \underset{u \sim U(\partial B_1(0))}{\mathbb{E}}[f_t(x + \delta A_t u)], \quad \widehat{g}_t = \frac{d}{\delta} f_t(y_t) A_t^{-1} u_t, \quad y_t = x_t + \delta A_t u_t$$

*for arbitrary $A_t$, $\delta$, and $u_t$. Let $x^* = \operatorname{argmin}_{x \in \mathcal{K}} \sum_{t=1}^T f_t(x)$, and let $x_\epsilon^* \in \operatorname{argmin}_{y \in \mathcal{K}, dist(y, \partial \mathcal{K}) > \epsilon} \|y - x^*\|$. Assume that we play $y_t$ at every round. Then the following structural estimate holds:*

$$\operatorname{Reg}_T(\mathcal{A}) = \mathbb{E}[\sum_{t=1}^T f_t(y_t) - f_t(x^*)] \le \epsilon L T + 2L\delta DT + \sum_{t=1}^T \mathbb{E}[\widehat{f}_t(x_t) - \widehat{f}_t(x_\epsilon^*)].$$

Thus, at the price of $\epsilon L T + 2L\delta DT$, it suffices to look at the performance of the averaged losses for the algorithm. Notice that the only assumptions we have made so far are that we play points sampled on an ellipsoid around the desired point scaled by $\delta$ and that the loss functions are Lipschitz.

**Lemma 6** ($\mathcal{C}^{0,1}$ Structural bound on smoothed losses in terms of averaged losses). *Let $(f_t)_{t=1}^T$ be a sequence of loss functions, and assume that $f_t \colon \mathcal{K} \to \mathbb{R}_+$ is $C$-bounded and $L$-Lipschitz, where $\mathcal{K} \subset \mathbb{R}^d$. Denote*

$$\widehat{f}_t(x) = \underset{u \sim U(\partial B_1(0))}{\mathbb{E}}[f_t(x + \delta A_t u)], \quad \widehat{g}_t = \frac{d}{\delta} f_t(y_t) A_t^{-1} u_t, \quad y_t = x_t + \delta A_t u_t$$

*for arbitrary $A_t$, $\delta$, and $u_t$. Let $x^* = \operatorname{argmin}_{x \in \mathcal{K}} \sum_{t=1}^T f_t(x)$, and let $x_\epsilon^* \in \operatorname{argmin}_{y \in \mathcal{K}, dist(y, \partial \mathcal{K}) > \epsilon} \|y - x^*\|$. Furthermore, denote*

$$\bar{f}_t(x) = \frac{1}{k+1} \sum_{i=0}^k \widehat{f}_{t-i}(x), \quad \bar{g}_t = \frac{1}{k+1} \sum_{i=0}^k \widehat{g}_{t-i}.$$

*Assume that we play $y_t$ at every round. Then we have the structural estimate:*

$$\sum_{t=1}^T \mathbb{E}\left[\widehat{f}_t(x_t) - \widehat{f}_t(x_\epsilon^*)\right] \le \frac{Ck}{2} + LT \sup_{t \in [1,T], i \in [0, k \wedge t]} \mathbb{E}[\|x_{t-i} - x_t\|_2] + \sum_{t=1}^T \mathbb{E}\left[\bar{g}_t^\top (x_t - x_\epsilon^*)\right].$$

While we use averaged smoothed losses as in [Dekel et al., 2015], the analysis in this lemma is actually somewhat different. Because Dekel et al. [2015] always assume that the loss functions are in $\mathcal{C}^{1,1}$, they elect to use the following decomposition:

$$\widehat{f}_t(x_t) - \widehat{f}_t(x_\epsilon^*) = \widehat{f}_t(x_t) - \bar{f}_t(x_t) + \bar{f}_t(x_t) - \bar{f}_t(x_\epsilon^*) + \bar{f}_t(x_\epsilon^*) - \widehat{f}_t(x_\epsilon^*).$$

This is because they can relate $\nabla \bar{f}_t(x) = \frac{1}{k+1} \sum_{i=0}^k \nabla \widehat{f}_{t-i}(x_\epsilon)$ to $\bar{g}_t = \frac{1}{k+1} \sum_{i=0}^k \nabla \widehat{f}_{t-i}(x_{t-i})$ using the fact that the gradients are Lipschitz. Since the gradients of $\mathcal{C}^{0,1}$ functions are not Lipschitz, we cannot use the same analysis. Instead, we use the decomposition

$$\widehat{f}_t(x_t) - \widehat{f}_t(x_\epsilon^*) = \widehat{f}_t(x_t) - \widehat{f}_{t-i}(x_{t-i}) + \widehat{f}_{t-i}(x_{t-i}) - \bar{f}_t(x_\epsilon^*) + \bar{f}_t(x_\epsilon^*) - \widehat{f}_t(x_\epsilon^*).$$

The next lemma affirms that we do indeed get the improved $\frac{1}{(k+1)^2}$ factor from predicting the predictable component of the average gradient.

**Lemma 7** ($\mathbb{C}^{0,1}$ Algorithmic bound on the averaged losses). *Let $(f_t)_{t=1}^T$ be a sequence of loss functions, and assume that $f_t \colon \mathcal{K} \to \mathbb{R}_+$ is $C$-bounded and $L$-Lipschitz, where $\mathcal{K} \subset \mathbb{R}^d$. Let $x^* = \operatorname{argmin}_{x \in \mathcal{K}} \sum_{t=1}^T f_t(x)$, and let $x_\epsilon^* \in \operatorname{argmin}_{y \in \mathcal{K}, dist(y, \partial \mathcal{K}) > \epsilon} \|y - x^*\|$. Assume that we play according to the algorithm with $\eta k \leq \frac{\delta}{12Cd}$. Then we maintain the following guarantee:*

$$\sum_{t=1}^T \mathbb{E}\left[\bar{g}_t^\top (x_t - x_\epsilon^*)\right] \leq \frac{2Cd^2\eta T}{\delta^2(k+1)^2} + \frac{1}{\eta}\mathcal{R}(x_\epsilon^*).$$

So far, we have demonstrated a bound on the regret of the form:

$$\operatorname{Reg}_T(\mathcal{A}) \leq \epsilon LT + 2L\delta DT + \frac{Ck}{2} + LT \sup_{t \in [T], i \in [k \wedge t]} \mathbb{E}[\|x_{t-i} - x_t\|_2] + \frac{2Cd^2\eta T}{\delta^2(k+1)^2} + \frac{1}{\eta}\mathcal{R}(x_\epsilon).$$

Thus, it remains to find a tight bound on $\sup_{t \in [1,T], i \in [0,k \wedge t]} \mathbb{E}[\|x_{t-i} - x_t\|_2]$, which measures the stability of the actions across the history that we average over. This result is similar to that of Dekel et al. [2015], except that we additionally need to account for the optimistic gradient prediction used.

**Lemma 8** ($\mathbb{C}^{0,1}$ Algorithmic bound on the stability of actions). *Let $(f_t)_{t=1}^T$ be a sequence of loss functions, and assume that $f_t \colon \mathcal{K} \to \mathbb{R}_+$ is $C$-bounded and $L$-Lipschitz, where $\mathcal{K} \subset \mathbb{R}^d$. Assume that we play according to the algorithm with $\eta k \leq \frac{\delta}{12Cd}$. Then the following estimate holds:*

$$\mathbb{E}[\|x_{t-i} - x_t\|_2] \leq 2\eta k D \left(\frac{\sqrt{3}L^{1/2}}{k} + \sqrt{2}DL + \frac{\sqrt{48}Cd}{\sqrt{k}\delta}\right).$$

*Proof.* [of Theorem 1] Putting all the pieces together from Lemmas 5, 6, 7, 8, shows that

$$\operatorname{Reg}_T(\mathcal{A}) \leq \epsilon LT + L\delta DT + \frac{Ck}{2} + \frac{2Cd^2\eta T}{\delta^2(k+1)^2} + \frac{1}{\eta}\mathcal{R}(x_\epsilon) + LT2\eta D\left[\sqrt{3}L^{1/2} + \sqrt{2}DLk + \frac{\sqrt{48}Cd\sqrt{k}}{\delta}\right].$$

Since $x_\epsilon$ is at least $\epsilon$ away from the boundary, it follows from [Abernethy and Rakhlin, 2009] that $\mathcal{R}(x_\epsilon) \leq \nu \log(1/\epsilon)$. Plugging in the stated quantities for $\eta$, $k$, and $\delta$ yields the result. $\square$

## 5.3 $\mathbb{C}^{1,1}$ analysis

The analysis of the $\mathbb{C}^{1,1}$ regret bound is similar to the $\mathbb{C}^{0,1}$ case. The only difference is that we leverage the higher regularity of the losses to provide a more refined estimate on the cost of approximating $f_t$ with $\widehat{f}_t$. Apart from that, we will reuse the bounds derived in Lemmas 6, 7, and 8. The proof of the following lemma, along with that of Theorem 2, is provided in Appendix 7.3.

**Lemma 9** ($\mathbb{C}^{1,1}$ Structural bound on true losses in terms of smoothed losses). *Let $(f_t)_{t=1}^T$ be a sequence of loss functions, and assume that $f_t \colon \mathcal{K} \to \mathbb{R}_+$ is $C$-bounded, $L$-Lipschitz, and $H$-smooth, where $\mathcal{K} \subset \mathbb{R}^d$. Denote*

$$\widehat{f}_t(x) = \mathbb{E}_{u \sim U(\partial B_1(0))}[f_t(x + \delta A_t u)], \quad \widehat{g}_t = \frac{d}{\delta}f_t(y_t)A_t^{-1}u_t, \quad y_t = x_t + \delta A_t u_t$$

*for arbitrary $A_t$, $\delta$, and $u_t$. Let $x^* = \operatorname{argmin}_{x \in \mathcal{K}} \sum_{t=1}^T f_t(x)$, and let $x_\epsilon^* \in \operatorname{argmin}_{y \in \mathcal{K}, dist(y, \partial \mathcal{K}) > \epsilon} \|y - x^*\|$. Assume that we play $y_t$ at every round. Then the following structural estimate holds:*

$$\operatorname{Reg}_T(\mathcal{A}) = \mathbb{E}[\sum_{t=1}^T f_t(y_t) - f_t(x^*)] \leq \epsilon LT + 2H\delta^2 D^2 T + \sum_{t=1}^T \mathbb{E}[\widehat{f}_t(x_t) - \widehat{f}_t(x_\epsilon^*)].$$

# 6 Conclusion

We designed a computationally efficient algorithm for bandit convex optimization admitting state-of-the-art guarantees for $\mathbb{C}^{0,1}$ and $\mathbb{C}^{1,1}$ loss functions. This was achieved using the general and powerful technique of *predicting* predictable information re-use. The ideas we describe here are directly applicable to other areas of optimization, in particular stochastic optimization.

**Acknowledgements**

This work was partly funded by NSF CCF-1535987 and IIS-1618662 and NSF GRFP DGE-1342536.

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
