[Supplementary Material]

# 7 Appendix

## 7.1 Properties of self-concordant barriers

This section highlights some properties of self-concordant barriers that will be useful in this work.

The ellipsoid induced by a self-concordant barrier at each point in the interior of the feasible set, $\{y \in \mathbb{R}^d \colon \|y\|_x \leq 1\}$, is called the Dikin ellipsoid. The first result tells us that we can sample around the Dikin ellipsoid without worrying about leaving the feasible region.

**Theorem 3** (Theorem 2.1.1 [Nesterov and Nemirovskii, 1994])**.** *Let $\mathcal{K}$ be a closed convex set in $\mathbb{R}^d$, and let $\mathcal{R}$ be a $\nu$-self-concordant barrier for $\mathcal{K}$. Then for any $x \in \operatorname{int}(\mathcal{K})$, we have that $\{y \in \mathbb{R}^d \colon \|y - x\|_x < 1\} \subset \operatorname{int}(\mathcal{K})$.*

The next result, presented in [Dekel et al., 2015], is a variant of John's ellipsoid theorem for ellipsoids induced by self-concordant barriers. It shows that the Euclidean norm and the norm induced by the barrier are equivalent up to the diameter of the convex set.

**Lemma 10** (Lemma 6 [Dekel et al., 2015])**.** *Let $\mathcal{K}$ be a closed convex set in $\mathbb{R}^d$, and let $\mathcal{R}$ be a $\nu$-self-concordant barrier for $\mathcal{K}$. Then for any $x \in \mathcal{K}$ and $y \in \mathbb{R}^d$, the following inequality holds: $D^{-1}\|z\|_{x,*} \leq \|z\|_2 \leq D\|z\|_x$.*

The second result shows that the Hessian of a self-concordant barrier changes slowly within the Dikin ellipsoid of a point.

**Theorem 4** (Theorem 2.1.1 [Nesterov and Nemirovskii, 1994])**.** *Let $\mathcal{K}$ be a closed convex set in $\mathbb{R}^d$, and let $\mathcal{R}$ be a $\nu$-self-concordant barrier for $\mathcal{K}$. Then for any $x \in \operatorname{int}(\mathcal{K})$ and $z \in \mathbb{R}^d$ with $\|y\|_x \leq 1$, we have that*

$$\left(1 - \|y\|_x\right)^2 \nabla^2 \mathcal{R}(x) \preccurlyeq \nabla^2 \mathcal{R}(x + y) \preccurlyeq \left(1 - \|y\|_x\right)^{-2} \nabla^2 \mathcal{R}(x).$$

The next result tells us that outside of an $\epsilon$ annulus at the boundary of $\mathcal{K}$, a self-concordant barrier grows at most logarithmically.

**Proposition 1** (Proposition 2.3.2 [Nesterov and Nemirovskii, 1994])**.** *Let $\mathcal{K}$ be a closed convex set in $\mathbb{R}^d$, and let $\mathcal{R}$ be a $\nu$-self-concordant barrier for $\mathcal{K}$. For any $\epsilon \in (0, 1]$, let $K_{y,\epsilon} = \{y + (1 - \epsilon)(x - y) \colon x \in \mathcal{K}\}$. Then for all $x \in \mathcal{K}_{y,\epsilon}$, the following inequality holds: $R(x) \leq \nu \log(1/\epsilon)$.*

## 7.2 Proofs for $\mathcal{C}^{0,1}$ analysis

**Lemma 5** ($\mathcal{C}^{0,1}$ Structural bound on true losses in terms of smoothed losses)**.** *Let $(f_t)_{t=1}^T$ be a sequence of loss functions, and assume that $f_t \colon \mathcal{K} \to \mathbb{R}_+$ is C-bounded and L-Lipschitz, where $\mathcal{K} \subset \mathbb{R}^d$. Denote*

$$\widehat{f}_t(x) = \mathop{\mathbb{E}}_{u \sim U(\partial B_1(0))}[f_t(x + \delta A_t u)], \quad \widehat{g}_t = \frac{d}{\delta} f_t(y_t) A_t^{-1} u_t, \quad y_t = x_t + \delta A_t u_t$$

*for arbitrary $A_t$, $\delta$, and $u_t$. Let $x^* = \operatorname{argmin}_{x \in \mathcal{K}} \sum_{t=1}^T f_t(x)$, and let $x_\epsilon^* \in \operatorname{argmin}_{y \in \mathcal{K}, dist(y, \partial \mathcal{K}) > \epsilon} \|y - x^*\|$. Assume that we play $y_t$ at every round. Then the following structural estimate holds:*

$$\operatorname{Reg}_T(\mathcal{A}) = \mathbb{E}[\sum_{t=1}^T f_t(y_t) - f_t(x^*)] \leq \epsilon L T + 2L\delta D T + \sum_{t=1}^T \mathbb{E}[\widehat{f}_t(x_t) - \widehat{f}_t(x_\epsilon^*)].$$

*Proof.* Then using that the losses are Lipschitz, that $\widehat{f}_t(x) \geq f_t(x)$, and that we sample around ellipsoids scaled by $\delta$,

$$
\begin{aligned}
\text{Reg}_T(\mathcal{A}) &= \mathbb{E}[\sum_{t=1}^{T} f_t(y_t) - f_t(x^*)] \\
&= \mathbb{E}[\sum_{t=1}^{T} f_t(y_t) - \widehat{f}_t(y_t) + \widehat{f}_t(y_t) - \widehat{f}_t(x_t) + \widehat{f}_t(x_t) - \widehat{f}_t(x_\epsilon^*) + \widehat{f}_t(x_\epsilon^*) - f_t(x_\epsilon^*) \\
&\quad + f_t(x_\epsilon^*) - f_t(x^*)] \\
&\leq \mathbb{E}[\sum_{t=1}^{T} \widehat{f}_t(y_t) - \widehat{f}_t(x_t) + \widehat{f}_t(x_\epsilon^*) - f_t(x_\epsilon^*)] + \sum_{t=1}^{T} \mathbb{E}[\widehat{f}_t(x_t) - \widehat{f}_t(x_\epsilon^*)] + \epsilon LT \\
&\leq 2L\delta DT + \epsilon LT + \sum_{t=1}^{T} \mathbb{E}[\widehat{f}_t(x_t) - \widehat{f}_t(x_\epsilon^*)].
\end{aligned}
$$
$\qquad\square$

**Lemma 6** ($\mathcal{C}^{0,1}$ Structural bound on smoothed losses in terms of averaged losses). *Let $(f_t)_{t=1}^T$ be a sequence of loss functions, and assume that $f_t \colon \mathcal{K} \to \mathbb{R}_+$ is C-bounded and L-Lipschitz, where $\mathcal{K} \subset \mathbb{R}^d$. Denote*

$$
\widehat{f}_t(x) = \mathop{\mathbb{E}}_{u \sim U(\partial B_1(0))}[f_t(x + \delta A_t u)], \quad \widehat{g}_t = \frac{d}{\delta} f_t(y_t) A_t^{-1} u_t, \quad y_t = x_t + \delta A_t u_t
$$

*for arbitrary $A_t$, $\delta$, and $u_t$. Let $x^* = \operatorname{argmin}_{x \in \mathcal{K}} \sum_{t=1}^T f_t(x)$, and let $x_\epsilon^* \in \operatorname{argmin}_{y \in \mathcal{K}, \text{dist}(y, \partial\mathcal{K}) > \epsilon} \|y - x^*\|$. Furthermore, denote*

$$
\bar{f}_t(x) = \frac{1}{k+1} \sum_{i=0}^{k} \widehat{f}_{t-i}(x), \quad \bar{g}_t = \frac{1}{k+1} \sum_{i=0}^{k} \widehat{g}_{t-i}.
$$

*Assume that we play $y_t$ at every round. Then we have the structural estimate:*

$$
\sum_{t=1}^{T} \mathbb{E}\left[\widehat{f}_t(x_t) - \widehat{f}_t(x_\epsilon^*)\right] \leq \frac{Ck}{2} + LT \sup_{t \in [1,T], i \in [0, k \wedge t]} \mathbb{E}[\|x_{t-i} - x_t\|_2] + \sum_{t=1}^{T} \mathbb{E}\left[\bar{g}_t^\top (x_t - x_\epsilon^*)\right].
$$

*Proof.* The following decomposition holds:

$$
\begin{aligned}
\sum_{t=1}^{T} \mathbb{E}\left[\widehat{f}_t(x_t) - \widehat{f}_t(x_\epsilon^*)\right] &= \sum_{t=1}^{T} \mathbb{E}\left[\frac{1}{k+1} \sum_{i=0}^{k} \left(\widehat{f}_t(x_t) - \widehat{f}_{t-i}(x_{t-i})\right)\right. \\
&\quad \left. + \frac{1}{k+1} \sum_{i=0}^{k} \left(\widehat{f}_{t-i}(x_{t-i}) - \bar{f}_t(x_\epsilon^*)\right) + \frac{1}{k+1} \left(\sum_{i=0}^{k} \bar{f}_t(x_\epsilon) - \widehat{f}_t(x_\epsilon^*)\right)\right] \\
&= \sum_{t=1}^{T} \mathbb{E}[(i) + (ii) + (iii)].
\end{aligned}
$$

For the first term (i), we have the estimate

$$\sum_{t=1}^{T} \frac{1}{k+1} \sum_{i=0}^{k} \left( \widehat{f}_t(x_t) - \widehat{f}_{t-i}(x_{t-i}) \right) = \sum_{t=T-k+1}^{T} \frac{1}{k+1} \sum_{i=0}^{k} \widehat{f}_t(x_t) - \widehat{f}_{t-i}(x_{t-i})$$

$$= \sum_{t=T-k+1}^{T} \frac{1}{k+1} \left( k - [k - (t - (T-k))] \right) \widehat{f}_t(x_t)$$

$$= \sum_{t=T-k+1}^{T} \frac{1}{k+1} (t - T + k) \widehat{f}_t(x_t)$$

(if $f \leq C$, then $\widehat{f} \leq C$)

$$\leq \sum_{t=T-k+1}^{T} \frac{1}{k+1} (t - T + k) C$$

$$= \sum_{t=1}^{k-1} \frac{1}{k+1} (t + T - k - T + k) C$$

$$= \sum_{t=1}^{k-1} \frac{1}{k+1} t C = \frac{(k-1)k}{2(k+1)} C \leq \frac{k}{2} C.$$

For the third term (iii), we can say that

$$\frac{1}{k+1} \left( \sum_{i=0}^{k} \bar{f}_t(x_\epsilon^*) - \widehat{f}_t(x_\epsilon^*) \right) = \sum_{t=1}^{T} \frac{1}{k+1} \sum_{i=0}^{k} \widehat{f}_{t-i}(x_\epsilon^*) - \widehat{f}_t(x_\epsilon^*)$$

$$= \sum_{t=k+1}^{T} \frac{1}{k+1} \sum_{i=0}^{k} \widehat{f}_{t-i}(x_\epsilon^*) - \widehat{f}_t(x_\epsilon^*)$$

$$+ \sum_{t=1}^{k} \frac{1}{k+1} \sum_{i=0}^{k} \widehat{f}_{t-i}(x_\epsilon^*) - \widehat{f}_t(x_\epsilon^*),$$

where the first term is equal to 0 and the second term is $\leq 0$ because $f \geq 0$. Finally, for the second term (ii), we have that

$$\mathbb{E}\left[ \sum_{t=1}^{T} \frac{1}{k+1} \sum_{i=0}^{k} \widehat{f}_{t-i}(x_{t-i}) - \bar{f}_t(x_\epsilon^*) \right] = \mathbb{E}\left[ \sum_{t=1}^{T} \frac{1}{k+1} \sum_{i=0}^{k} \widehat{f}_{t-i}(x_{t-i}) - \widehat{f}_{t-i}(x_\epsilon^*) \right]$$

$$\leq \mathbb{E}\left[ \sum_{t=1}^{T} \frac{1}{k+1} \sum_{i=0}^{k} \widehat{g}_{t-i}^{\top}(x_{t-i} - x_\epsilon^*) \right]$$

$$= \mathbb{E}\left[ \sum_{t=1}^{T} \frac{1}{k+1} \sum_{i=0}^{k} \widehat{g}_{t-i}^{\top}(x_t - x_\epsilon^*) + \widehat{g}_{t-i}^{\top}(x_{t-i} - x_t) \right]$$

$$= \mathbb{E}\left[ \sum_{t=1}^{T} \bar{g}_t^{\top}(x_t - x_\epsilon^*) + \frac{1}{k+1} \sum_{i=0}^{k} \widehat{g}_{t-i}^{\top}(x_{t-i} - x_t) \right].$$

Next, by the linearity of expectation, we can write

$$\mathbb{E}\left[\sum_{t=1}^{T} \frac{1}{k+1} \sum_{i=0}^{k} \widehat{f}_{t-i}(x_{t-i}) - \bar{f}_t(x_\epsilon^*)\right]$$

$$= \sum_{t=1}^{T} \mathbb{E}\left[\bar{g}_t^\top (x_t - x_\epsilon^*)\right] + \sum_{t=1}^{T} \frac{1}{k+1} \sum_{i=0}^{k} \mathbb{E}\left[\widehat{g}_{t-i}^\top (x_{t-i} - x_t)\right]$$

$$= \sum_{t=1}^{T} \mathbb{E}\left[\bar{g}_t^\top (x_t - x_\epsilon^*)\right] + \sum_{t=1}^{T} \frac{1}{k+1} \sum_{i=0}^{k} \mathbb{E}\left[\nabla \hat{f}_{t-i}(x_{t-i})^\top (x_{t-i} - x_t)\right]$$

$$\leq \sum_{t=1}^{T} \mathbb{E}\left[\bar{g}_t^\top (x_t - x_\epsilon^*)\right] + \sum_{t=1}^{T} \frac{1}{k+1} \sum_{i=0}^{k} \mathbb{E}\left[\|\nabla \hat{f}_{t-i}(x_{t-i})\|_2 \|x_{t-i} - x_t\|_2\right]$$

$$\leq \sum_{t=1}^{T} \mathbb{E}\left[\bar{g}_t^\top (x_t - x_\epsilon^*)\right] + LT \sup_{t\in[1,T], i\in[0,k\wedge t]} \mathbb{E}[\|x_{t-i} - x_t\|_2]. \qquad \square$$

**Lemma 7** ($\mathcal{C}^{0,1}$ Algorithmic bound on the averaged losses). *Let $(f_t)_{t=1}^{T}$ be a sequence of loss functions, and assume that $f_t\colon \mathcal{K} \to \mathbb{R}_+$ is $C$-bounded and $L$-Lipschitz, where $\mathcal{K} \subset \mathbb{R}^d$. Let $x^* = \operatorname{argmin}_{x\in\mathcal{K}} \sum_{t=1}^{T} f_t(x)$, and let $x_\epsilon^* \in \operatorname{argmin}_{y\in\mathcal{K}, dist(y,\partial\mathcal{K})>\epsilon} \|y - x^*\|$. Assume that we play according to the algorithm with $\eta k \leq \frac{\delta}{12Cd}$. Then we maintain the following guarantee:*

$$\sum_{t=1}^{T} \mathbb{E}\left[\bar{g}_t^\top (x_t - x_\epsilon^*)\right] \leq \frac{2Cd^2\eta T}{\delta^2(k+1)^2} + \frac{1}{\eta}\mathcal{R}(x_\epsilon^*).$$

*Proof.* The first part of the proof is very similar to the analysis given in [Rakhlin and Sridharan, 2013]. For completeness and ease of presentation, we present the full argument here.

Our algorithm is based on the update rule:

$$x_{t+1} = \operatorname*{argmin}_{x} \eta(\bar{g}_{1:t} + \widetilde{g}_{t+1})^\top x + \mathcal{R}(x),$$

where

$$\widetilde{g}_{t+1} = \frac{1}{k+1} \sum_{i=0}^{k-1} \widehat{g}_{t-i} = \frac{1}{k+1} \sum_{i=0}^{k} \widehat{g}_{t+1-i} - \frac{1}{k+1} \widehat{g}_{t+1}.$$

Let $z_t = \operatorname{argmin}_x \eta(\bar{g}_{1:t})^\top x + \mathcal{R}(x)$. Then

$$\sum_{t=1}^{T} \mathbb{E}[\bar{g}_t^\top (x_t - x)] = \sum_{t=1}^{T} \bar{g}_t^\top (x_t - z_t) + \bar{g}_t^\top (z_t - x)$$

$$= \sum_{t=1}^{T} (\bar{g}_t - \widetilde{g}_t)^\top (x_t - z_t) + \widetilde{g}_t^\top (x_t - z_t) + \bar{g}_t^\top (z_t - x)$$

Now we want to show that $\forall x$,

$$\sum_{t=1}^{T} \widetilde{g}_t^\top (x_t - z_t) + \bar{g}_t^\top z_t \leq \frac{1}{\eta}\mathcal{R}(x) + \sum_{t=1}^{T} \bar{g}_t^\top x.$$

For $T = 1$, $\forall x$, we need to show that

$$\widetilde{g}_1^\top (x_1 - y_1) + \bar{g}_1^\top y_1 \leq \frac{1}{\eta}\mathcal{R}(x) + \bar{g}_1^\top x.$$

But $\widetilde{g}_1 = 0$, and so the result follows from the definition of $y_1$.

Now assume that the result is true for $T - 1$:

$$\sum_{t=1}^{T} \widetilde{g}_t^\top (x_t - z_t) + \bar{g}_t^\top z_t = \sum_{t=1}^{T-1} \widetilde{g}_t^\top (x_t - z_t) + \bar{g}_t^\top z_t + \widetilde{g}_T^\top (x_t - z_t) + \bar{g}_T^\top y_T$$

$$\leq \frac{1}{\eta} \mathcal{R}(x_T) + \sum_{t=1}^{T-1} \bar{g}_t^\top x_T + \widetilde{g}_T^\top (x_T - y_T) + \bar{g}_T \cdot y_T$$

(by the induction hypothesis)

$$= \frac{1}{\eta} \mathcal{R}(x_T) + \left( \sum_{t=1}^{T-1} \bar{g}_t + \widetilde{g}_T \right)^\top x_T - \widetilde{g}_T^\top y_T + \bar{g}_T^\top y_T$$

$$\leq \frac{1}{\eta} \mathcal{R}(y_T) + \left( \sum_{t=1}^{T-1} \bar{g}_t + \widetilde{g}_T \right)^\top y_T - \widetilde{g}_T^\top y_T + \bar{g}_T^\top y_T$$

(by definition of $x_T$)

$$= \frac{1}{\eta} \mathcal{R}(y_T) + \left( \sum_{t=1}^{T} \bar{g}_t \right)^\top y_T$$

$$\leq \frac{1}{\eta} \mathcal{R}(x) + \left( \sum_{t=1}^{T} \bar{g}_t \right)^\top x \quad \forall x$$

(by definition of $y_T$).

Thus, we have that

$$\sum_{t=1}^{T} (\bar{g}_t - \widetilde{g}_t)^\top (x_t - z_t) + \widetilde{g}_t^\top (x_t - z_t) + \bar{g}_t^\top (z_t - x) \leq \sum_{t=1}^{T} (\bar{g}_t - \widetilde{g}_t)^\top (x_t - z_t) + \frac{1}{\eta} \mathcal{R}(x).$$

Now, we can use duality to apply the bound:

$$\sum_{t=1}^{T} (\bar{g}_t - \widetilde{g}_t)^\top (x_t - z_t) + \frac{1}{\eta} \mathcal{R}(x) \leq \sum_{t=1}^{T} \|\bar{g}_t - \widetilde{g}_t\|_{x_t,*} \|x_t - z_t\|_{x_t} + \frac{1}{\eta} \mathcal{R}(x)$$

and we want to show that

$$x_{t+1} = \operatorname*{argmin}_{x} (\bar{g}_{1:t} + \widetilde{g}_{t+1})^\top x + \frac{1}{\eta} \mathcal{R}(x)$$

$$y_{t+1} = \operatorname*{argmin}_{x} \bar{g}_{1:t+1}^\top x + \frac{1}{\eta} \mathcal{R}(x)$$

imply that

$$\|x_t - z_t\|_{x_t} \leq 2\eta \|\bar{g}_t - \widetilde{g}_t\|_{x_t,*}.$$

Toward this end, we recall the following result from [Nesterov, 2004]:

**Theorem 5** (Theorem 2.2 [Nesterov, 2004])**.** *Let* $\lambda(x, F) = \|\nabla F(x)\|_{\nabla^2 F(x)^{-1}} = \|\nabla F(x)\|_{x,*}$ *be the Newton decrement of* $F$ *at* $x$. *Suppose that* $\lambda(x, F) \leq \frac{1}{2}$ *and* $F$ *is a self-concordant barrier. Then*

$$\|x - \operatorname{argmin} F\|_x \leq 2\lambda(x, F).$$

Using this result, consider $F_{t+1}(x) = \bar{g}_{1:t+1}^\top x + \frac{1}{\eta} \mathcal{R}(x)$. Then the theorem implies that if $\lambda(x_t, F_t) \leq \frac{1}{2}$ (which is true if $\eta k \leq \frac{\delta}{12Cd}$), then

$$\|x_t - z_t\|_{x_t} = \|x_t - \operatorname{argmin} F_t\|_{x_t} \leq 2\lambda(x_t, F_t) = 2\eta \|\bar{g}_t - \widetilde{g}_t\|_{x_t,*}.$$

Thus, we have shown that

$$\sum_{t=1}^{T} \mathbb{E}[\bar{g}_t^\top (x_t - x)] \leq \frac{1}{\eta} \mathcal{R}(x) + \sum_{t=1}^{T} 2\eta \, \mathbb{E}[\|\bar{g}_t - \widetilde{g}_t\|_{x_t,*}^2]$$

We can also estimate that

$$\mathbb{E}[\|\bar{g}_t - \widetilde{g}_t\|_{x_t,*}^2] = \mathbb{E}[\|\frac{1}{k+1}\sum_{i=0}^{k}\widehat{g}_{t-i} - \sum_{i=1}^{k}\widehat{g}_{t-i}\|_{x_t,*}^2 = \mathbb{E}[\|\widehat{g}_t\|_{x_t,*}^2 \frac{1}{(k+1)^2} \leq \frac{Cd^2}{\delta^2(k+1)^2}$$

$\square$

**Lemma 8** ($\mathcal{C}^{0,1}$ Algorithmic bound on the stability of actions). *Let $(f_t)_{t=1}^T$ be a sequence of loss functions, and assume that $f_t\colon \mathcal{K} \to \mathbb{R}_+$ is $C$-bounded and $L$-Lipschitz, where $\mathcal{K} \subset \mathbb{R}^d$. Assume that we play according to the algorithm with $\eta k \leq \frac{\delta}{12Cd}$. Then we have the following estimate:*

$$\mathbb{E}[\|x_{t-i} - x_t\|_2] \leq 2\eta k D \left(\frac{\sqrt{3}L^{1/2}}{k} + \sqrt{2}DL + \frac{\sqrt{48}Cd}{\sqrt{k}\delta}\right)$$

*Proof.* As a first step, we can write that

$$\mathbb{E}[\|x_{t-i} - x_t\|_2] \leq \sum_{s=t-i}^{t-1} \mathbb{E}[\|x_s - x_{s+1}\|_2].$$

Now, Lemma 10 informs us that we can switch between the local norms and the Euclidean norm while only paying a price of $D$. This allows us to say that $\|x_s - x_{s+1}\|_2 \leq D\|x_s - x_{s+1}\|_{x_s}$.

Let $G_{t+1}(x) = \eta(\bar{g}_{1:t} + \widetilde{g}_{t+1})^\top x + \mathcal{R}(x)$. If $\eta > 0$ is such that $\lambda(x_t, G_{t+1}) \leq \frac{1}{4}$, then Theorem 2.2 of Nesterov implies that

$$\|x_s - x_{s+1}\|_{x_s} = \|x_s - \arg\min G_{t+1}\|_{x_s} \leq 2\lambda(x_s, G_{s+1}).$$

Since

$$\nabla G_{t+1}(x_t) = \nabla G_t(x_t) + \eta(\bar{g}_t + \widetilde{g}_{t+1} - \widetilde{g}_t) = \eta(\bar{g}_t + \widetilde{g}_{t+1} - \widetilde{g}_t),$$

we can write the Newton decrement as $\lambda(x_s, G_{s+1}) = \eta\|\bar{g}_s + \widetilde{g}_{s+1} - \widetilde{g}_s\|_{x_s,*}$.

By Jensen's inequality, we can write

$$\mathbb{E}[\|\bar{g}_t + \widetilde{g}_{t+1} - \widetilde{g}_t\|_{x_t,*}] \leq \sqrt{\mathbb{E}[\|\bar{g}_t + \widetilde{g}_{t+1} - \widetilde{g}_t\|_{x_t,*}^2]}.$$

Expanding out these terms, we can see that

$$\bar{g}_t + \widetilde{g}_{t+1} - \widetilde{g}_t = \frac{1}{k+1}\sum_{i=0}^{k}\widehat{g}_{t-i} + \frac{1}{k+1}\sum_{i=0}^{k-1}\widehat{g}_{t-i} - \frac{1}{k+1}\sum_{i=1}^{k}\widehat{g}_{t-i}$$

$$= \frac{1}{k+1}\sum_{i=0}^{k}\widehat{g}_{t-i} + \frac{1}{k+1}\left(\widehat{g}_t - \widehat{g}_{t-k}\right)$$

which implies that

$$\|\bar{g}_t + \widetilde{g}_{t+1} - \widetilde{g}_t\|_{x_t,*} \leq \frac{1}{k+1}\|\sum_{i=0}^{k-1}\widehat{g}_{t-i}\|_{x_t,*} + \frac{1}{k+1}\|\widehat{g}_t\|_{x_t,*}.$$

Let $\mathbb{E}_t$ denote the conditional expectation at time $t$, where we condition on all the randomization for the player up to time $t$. Then by using the fact that $(\alpha + \beta + \gamma)^2 \leq 3(\alpha^2 + \beta^2 + \gamma^2)$, we have

$$\mathbb{E}[\|\bar{g}_t + \widetilde{g}_{t+1} - \widetilde{g}_t\|_{x_t,*}^2] \leq \frac{3}{k^2}\|\sum_{i=0}^{k-1}\mathbb{E}_{t-i}[\widehat{g}_{t-i}]\|_{x_t,*}^2 + \frac{3}{k^2}\mathbb{E}[\|\sum_{i=0}^{k-1}\widehat{g}_{t-i} - \mathbb{E}_{t-i}[\widehat{g}_{t-i}]\|_{x_t,*}^2] + \frac{3}{k^2}L$$

$$\leq \frac{3}{k^2}L + 2D^2L^2 + \frac{3}{k^2}\mathbb{E}[\|\sum_{i=0}^{k-1}\widehat{g}_{t-i} - \mathbb{E}_{t-i}[\widehat{g}_{t-i}]\|_{x_t,*}^2].$$

Denote $h_{t-i} = \hat{g}_{t-i} - \mathbb{E}_{t-i}[\hat{g}_{t-i}]$. Then we can write this last term as

$$\frac{3}{k^2}\mathbb{E}[\|\sum_{i=0}^{k-1}\hat{g}_{t-i} - \underset{t-i}{\mathbb{E}}[\hat{g}_{t-i}]\|_{x_t,*}^2] = \frac{3}{k^2}\mathbb{E}[\|\sum_{i=0}^{k-1}h_{t-i}\|_{x_t,*}^2].$$

We now want to relate the norm $\|\cdot\|_{x_t,*}$ to the earliest norm $\|\cdot\|_{x_{t-k},*}$ in the batch. To do this, we will show the following two facts:

1. $\|\bar{g}_t + \tilde{g}_{t+1} - \tilde{g}_t\|_{x_t,*} \leq \frac{2d}{\delta}$

2. $\forall 0 \leq i \leq k$ such that $t - i \geq 1$, we have

$$\frac{1}{2}\|z\|_{x_{t-i},*} \leq \|z\|_{x_t,*} \leq 2\|z\|_{x_{t-i},*}.$$

Writing out the expressions, we have that

$$\bar{g}_t = \frac{1}{k+1}\sum_{i=0}^{k}\hat{g}_{t-i}, \quad \tilde{g}_{t+1} = \frac{1}{k+1}\sum_{i=0}^{k-1}\hat{g}_{t-i}, \quad \tilde{g}_t = \frac{1}{k+1}\sum_{i=1}^{k}\hat{g}_{t-i}.$$

For $t = 1$, we have

$$\bar{g}_1 + \tilde{g}_2 - \tilde{g}_1 = \frac{1}{k+1}\hat{g}_1 + \frac{1}{k+1}\hat{g}_1 = \frac{2}{k+1}\hat{g}_1$$

and $\|\hat{g}_s\|_{x_s,*} \leq \frac{Cd}{\delta} \forall s$, which together imply that

$$\|\bar{g}_1 + \tilde{g}_2 - \tilde{g}_1\|_{x_1,*} = \frac{2}{k+1}\|\hat{g}_1\|_{x_1,*} \leq \frac{2}{k+1}\frac{Cd}{\delta} \leq \frac{2Cd}{\delta}$$

Let $t > 1$. Then

$$\|\bar{g}_t + \tilde{g}_{t+1} - \tilde{g}_t\|_{x_t,*} \leq \frac{1}{k+1}\left(\sum_{i=0}^{k-1}\|\hat{g}_{t-i}\|_{x_t,*} + \|\hat{g}_t\|_{x_t,*}\right).$$

By the induction hypothesis,

$$\|\bar{g}_s + \tilde{g}_{s+1} - \tilde{g}_s\|_{x_s,*} \leq \frac{2Cd}{\delta}\forall s < t.$$

This implies that

$$\eta\|\bar{g}_s + \tilde{g}_{s+1} - \tilde{g}_s\|_{x_s,*} \leq \eta\frac{2Cd}{\delta}\forall s < t.$$

If we take $12k\eta Cd \leq \delta$, then

$$\eta\|\bar{g}_s + \tilde{g}_{s+1} - \tilde{g}_s\|_{x_s,*} = \lambda(x_s, G_{s+1}) \leq \frac{1}{4},$$

such that

$$\|x_{s+1} - x_s\|_{x_s} \leq 2\eta\|\bar{g}_s + \tilde{g}_{s+1} - \tilde{g}_s\|_{x_s,*} \leq 4\eta\frac{Cd}{\delta} \leq \frac{1}{3}. \quad \text{(since } 12k\eta Cd \leq \delta\text{)}$$

Thus, we have shown that

$$\|x_{s+1} - x_s\|_{x_s} \leq \frac{1}{3} < 1.$$

By Theorem 4, this means that

$$\left(1 - 4\eta\frac{d}{\delta}\right)^2 \nabla^2\mathcal{R}(x_s)^{-1} \preccurlyeq \nabla^2\mathcal{R}(x_{s+1})^{-1} \preccurlyeq \left(1 - 4\eta\frac{d}{\delta}\right)^{-2}\nabla^2\mathcal{R}(x_s)^{-1},$$

which implies the following relation on the induced norms: $\forall z$,

$$\left(1 - 4\eta\frac{d}{\delta}\right)\|z\|_{x_s,*} \leq \|z\|_{x_{s+1},*} \leq \left(1 - 4\eta\frac{d}{\delta}\right)^{-1}\|z\|_{x_s,*},$$

and recursively, $\forall s \in [t-k,t]$, $\forall z$,

$$\left(1 - 4\eta\frac{d}{\delta}\right)^k\|z\|_{x_s,*} \leq \|z\|_{x_t,*} \leq \left(1 - 4\eta\frac{d}{\delta}\right)^{-k}\|z\|_{x_s,*}.$$

Now, we want to show that $\left(1 - 4\eta\frac{d}{\delta}\right)^k \geq \frac{1}{2}$.

Since $12k\eta C\delta \leq \delta$, we have

$$4\eta\frac{Cd}{\delta} \leq \frac{1}{3} < \frac{1}{2}.$$

Using the fact that $1 - \beta \geq e^{-2\beta}$ for $\beta \in [0, \frac{1}{2}]$ implies that

$$\left[1 - (4\eta\frac{Cd}{\delta})\right]^k \geq e^{-8k\eta\frac{Cd}{\delta}} \geq e^{-2/3} \geq \frac{1}{2}.$$

Thus, we have that

$$\frac{1}{2}\|z\|_{x_s,*} \leq \|z\|_{x_t,*} \leq 2\|z\|_{x_s,*},$$

which is (2).

This also further implies that

$$\|\bar{g}_t + \widetilde{g}_{t+1} - \widetilde{g}_t\|_{x_t,*} \leq \frac{1}{k+1}\sum_{i=0}^{k-1}\|\widehat{g}_{t-i}\|_{x_t,*} + \frac{1}{k+1}\|\widehat{g}_t\|_{x_t,*}$$

$$\leq \frac{2}{k+1}\sum_{i=0}^{k-1}\|\widehat{g}_{t-i}\|_{x_{t-i},*} + \frac{1}{k+1}\|\widehat{g}_t\|_{x_t,*}$$

$$\leq \frac{2}{k+1}k\frac{Cd}{\delta} + \frac{1}{k+1}\frac{Cd}{\delta}$$

$$= \frac{1}{k+1}\frac{Cd}{\delta}(2k+1)$$

$$\leq \frac{2Cd}{\delta}.$$

Going back to the original quantity, we can now say that

$$\frac{3}{k^2}\mathbb{E}[\|\sum_{i=0}^{k-1}h_{t-i}\|^2_{x_t,*}] \leq \frac{3}{k^2}4\,\mathbb{E}[\|\sum_{i=0}^{k-1}h_{t-i}\|^2_{x_{t-k},*}]$$

$$= \frac{3}{k^2}4\sum_{i=0}^{k-1}\mathbb{E}[\|h_{t-i}\|^2_{x_{t-k},*}] \text{ (because } h_{t-i} \text{ is a martingale difference)}$$

$$\leq \frac{3}{k^2}16\sum_{i=0}^{k-1}\mathbb{E}[\|h_{t-i}\|^2_{x_{t-i},*}]$$

$$\leq \frac{3}{k^2}16\sum_{i=0}^{k-1}\mathbb{E}[\|\widehat{g}_{t-i}\|^2_{x_{t-i},*}]$$

$$\leq \frac{3}{k^2}16\sum_{i=0}^{k-1}\frac{C^2d^2}{\delta^2}$$

$$= \frac{3}{k}16\frac{C^2d^2}{\delta^2}.$$

$\square$

## 7.3 Proofs for $\mathcal{C}^{1,1}$ analysis

**Lemma 9** ($\mathcal{C}^{1,1}$ Structural bound on true losses in terms of smoothed losses). *Let $(f_t)_{t=1}^T$ be a sequence of loss functions, and assume that $f_t \colon \mathcal{K} \to \mathbb{R}_+$ is $C$-bounded, $L$-Lipschitz, and $H$-smooth, where $\mathcal{K} \subset \mathbb{R}^d$. Denote*

$$\widehat{f}_t(x) = \underset{u \sim U(\partial B_1(0))}{\mathbb{E}}[f_t(x + \delta A_t u)], \quad \widehat{g}_t = \frac{d}{\delta} f_t(y_t) A_t^{-1} u_t, \quad y_t = x_t + \delta A_t u_t$$

*for arbitrary $A_t$, $\delta$, and $u_t$. Let $x^* = \operatorname{argmin}_{x \in \mathcal{K}} \sum_{t=1}^T f_t(x)$, and let $x_\epsilon^* \in \operatorname{argmin}_{y \in \mathcal{K}, dist(y, \partial \mathcal{K}) > \epsilon} \|y - x^*\|$. Assume that we play $y_t$ at every round. Then we have the structural estimate:*

$$\operatorname{Reg}_T(\mathcal{A}) = \mathbb{E}[\sum_{t=1}^T f_t(y_t) - f_t(x^*)] \le \epsilon L T + 2 H \delta^2 D^2 T + \sum_{t=1}^T \mathbb{E}[\widehat{f}_t(x_t) - \widehat{f}_t(x_\epsilon^*)].$$

*Proof.* Then using the fact that the losses are Lipschitz and that we sample around ellipsoids scaled by $\delta$, we have that

$$\begin{aligned}
\operatorname{Reg}_T(\mathcal{A}) &= \mathbb{E}[\sum_{t=1}^T f_t(y_t) - f_t(x^*)] \\
&= \mathbb{E}[\sum_{t=1}^T f_t(y_t) - \widehat{f}_t(y_t) + \widehat{f}_t(y_t) - \widehat{f}_t(x_t) + \widehat{f}_t(x_t) - \widehat{f}_t(x_\epsilon^*) + \widehat{f}_t(x_\epsilon^*) - f_t(x_\epsilon^*) \\
&\quad + f_t(x_\epsilon^*) - f_t(x^*)] \\
&\le \mathbb{E}[\sum_{t=1}^T \widehat{f}_t(y_t) - \widehat{f}_t(x_t) + \widehat{f}_t(x_\epsilon^*) - f_t(x_\epsilon^*)] + \sum_{t=1}^T \mathbb{E}[\widehat{f}_t(x_t) - \widehat{f}_t(x_\epsilon^*)] + \epsilon L T \\
&\le 2 H \delta^2 D^2 T + \epsilon L T + \sum_{t=1}^T \mathbb{E}[\widehat{f}_t(x_t) - \widehat{f}_t(x_\epsilon^*)].
\end{aligned}$$

$\square$

**Theorem 2** ($\mathcal{C}^{1,1}$ Bound). *Let $\mathcal{K} \subset \mathbb{R}^d$ be a convex set with diameter $D$ and $(f_t)_{t=1}^T$ a sequence of loss functions with each $f_t \colon \mathcal{K} \to \mathbb{R}_+$ $C$-bounded, $L$-Lipschitz and $H$-smooth. Let $\mathcal{R}$ be a $\nu$-self-concordant barrier for $\mathcal{K}$. Then, for $\eta k \le \frac{\delta}{12 C d}$, the regret of* OPTIMISTICBCO *can be bounded as follows:*

$$\operatorname{Reg}_T(\text{OptimisticBCO}) \le \epsilon L T + H \delta^2 D^2 T$$
$$+ (TL + DHT) 2 \eta k D \left[ \frac{\sqrt{3} L^{1/2}}{k} + \sqrt{2} D L + \frac{\sqrt{48} C d}{\sqrt{k} \delta} \right] + \frac{1}{\eta} \log(1/\epsilon) + C k + \eta \frac{d^2 T}{\delta^2 (k+1)^2}.$$

*In particular, for $\eta = T^{-8/13} d^{-5/6}$, $\delta = T^{-5/26} d^{1/3}$, $k = T^{1/13} d^{5/3}$, the following guarantee holds for the regret of the algorithm:*

$$\operatorname{Reg}_T(\text{OptimisticBCO}) = \widetilde{\mathcal{O}}\left(T^{8/13} d^{5/3}\right).$$

*Proof.* Putting the pieces together from Lemmas 6, 7, 8, and 9, shows that

$$\begin{aligned}
\operatorname{Reg}_T(\mathcal{A}) &\le \epsilon L T + H \delta^2 D^2 T + \frac{C k}{2} + \frac{2 C d^2 \eta T}{\delta^2 (k+1)^2} + \frac{1}{\eta} \mathcal{R}(x_\epsilon) \\
&\quad + L T 2 \eta D \left[ \sqrt{3} L^{1/2} + \sqrt{2} D L k + \frac{\sqrt{48} C d \sqrt{k}}{\delta} \right]
\end{aligned}$$

Since $x_\epsilon$ is at least $\epsilon$ away from the boundary, it follows from [Abernethy and Rakhlin, 2009] that $\mathcal{R}(x_\epsilon) \le \nu \log(1/\epsilon)$.

Now, leaving only the $T$, $k$, $\eta$, $\delta$, and $\epsilon$ terms yields an expression of the form:

$$f(k, \eta, \delta, \epsilon) = \epsilon T + \frac{1}{\eta} \log(1/\epsilon) + \delta^2 T + k + \frac{\eta T d^2}{\delta^2 k^2} + T\eta \left[ 1 + k + \frac{k^{1/2} d}{\delta} \right]$$

Now, if we assume a priori that $k = \Omega(1)$ as $T \to \infty$ and take $\epsilon = T^{-1000}$ as in the statement, then we only care about the terms

$$\frac{1}{\eta} \log(1/\epsilon) + \delta^2 T + k + \frac{\eta T d^2}{\delta^2 k^2} + T\eta k + T\eta \frac{k^{1/2} d}{\delta}.$$

Plugging in the stated terms for $\eta$, $k$, and $\delta$ yields the result. $\qquad \square$