[Reviews · NeurIPS 2016]

Reviewer 1

Summary

This paper studies the online convex optimization in the bandit setting and improves over the state-of-the-art regret bounds in two cases: - It provides improved regret bound of T^{11/16} for Lipschitz functions and improves over the best known T^{5/6} regret bound - It attains T^{8/13} regret bound for smooth Lipschitz functions (Lipschitz gradients) and improves over the best known T^{2/3} regret bound.

Qualitative Assessment

The proposed algorithm combines the ideas from the smoothing and one-point gradient estimation techniques with self-concordant barriers to step away from boundary and the optimistic framework/gradual variation bounds with the partial averaging idea over a window proposed in a very recent work of Dekel et al. NIPS'15. The presentation of the paper was mostly clear. The claimed contributions are discussed in the light of existing results and the paper does survey related work appropriately. The paper is technically sound and the proofs seem to be correct as far as I checked. The improved regret bounds in this paper are interesting, but the paper lacks enough novelty. In particular, the proof of main results mostly relies on combining the existing methods, and is pretty straightforward following the analysis of Dekel et al. NIPS'15 and predictable sequences. However, I think the results by themselves are interesting enough to make paper worthy of publication.

Confidence in this Review

2-Confident (read it all; understood it all reasonably well)


Reviewer 2

Summary

The present paper provides a new algorithm---optimistic BCO---for bandit convex optimization. It leads to improved regret bounds among computationally efficient algorithms, for both Lipschitz loss functions and loss functions with Lipschitz gradients.

Qualitative Assessment

The presentation is globally clear, the paper seems technically solid, but a careful proofreading is necessary before publication. For instance there are too many notational inconsistencies throughout the paper. As far as originality is concerned, the improvement in the regret bounds is new, but they may seem incremental, especially because there are known to be suboptimal (putting algorithmic issues aside). Indeed Bubeck and Eldan (2015) proved through a non-constructive approach that the optimal rate in T is sqrt(T) up to log factors. The authors also mention the recent manuscript by Hazan and Li (2016) who claim to get the optimal sqrt(T) regret (up to log factors) but whose algorithm has an exponential dependency w.r.t. the dimension of the decision set, both in terms of its regret bound and its computational complexity. On the contrary, the algorithm provided here has a polynomial dependency w.r.t. the ambient dimension. I cannot figure out whether the prediction of information re- use is just a technical trick or if it can be used to get optimal bounds. Could the authors provide some insight about that? Besides, why just predict \bar{g}_{t+1} and not \bar{g}_{t+2},...,\bar{g}_{t+k}? Some specific comments: - Is it an oblivious adversary? (cf. definition of regret line 54) - Line 126: please specify \widehat{g}_{t-i} when t-i \leq 0. - Previous work : please also mention existing lower bounds - Figure 1: k is also a parameter? - \eta in the definition of x_{t+1} is not at the right place? See also line 183 and 330. - In theorem 1 and 2, must k be an integer? - In lemma 5, x is not defined, x^star= min ... - Line 218 missing square bracket - Line 317, please say somewhere that \widehat{f}_t(x) \geq f_t(x), - Line 330 I do not understand the conditional expectation, what is F_t? - Page 14: inversion between y_t and z_t and T and t - End of lemma 7 where do you check that \lambda(x_t,F_t)\leq 1/2? - Line 347 Since ... it is an x_t - Line 352 what is E_{t-i}? - Same line, I do not understand the upper bound of the third term by L. - Line 357 in Lemma 2 there is a factor C in the upper bound of the local norm of \widehat{g}_s.

Confidence in this Review

2-Confident (read it all; understood it all reasonably well)


Reviewer 3

Summary

The authors consider the fundamental online bandit convex optimization problem. They give improved regret algorithms with polynomial time complexity, that achieve regrets T^{11/16} (previous was T^{5/6}) for Lipschitz functions and T^{8/13} (previous was T^{2/3}) for Lipschitz functions with Lipschitz gradients. Better regret rates are only known existentially or via algorithms that require exponential computation. Their algorithm is based on the smoothing technique of Flaxman et al and Saha and Tewari, combined with the optimistic framework of Rakhlin and Shridharan. The algorithm is basically optimistic Follow the regularized leader, where the gradient used is the gradient of the smoothed function (smoothed with a uniform distribution in a delta-ball) and also averaged over a past recent window. The latter averaging over a window is an idea of a recent work of Dekel et al. 2015. The authors observe that if we also average over a recent window, then we essentially know the gradient of the next step, except for one term, which is the gradient that we are missing. Hence, we can "almost" predict the next step gradient. Thus we can use the optimistic framework of Rakhlin and Shridharan, which allows for the use of any such predictors and gives regret bounded by the prediction error. In this case, the latter will be of the order of the next step (non-averaged) gradient. This is the main idea that gives them an edge over the analysis in Dekel et al'15 and which gives the improved regret rates over Flaxman et al.

Qualitative Assessment

I found the paper very interesting. The BCO framework is very fundamental and the authors provide an improvement over the state of the art with a very simple algorithm that elegantly uses the optimistic learning framework of Rakhlin and Shridharan. For this reason I recommend acceptance.

Confidence in this Review

2-Confident (read it all; understood it all reasonably well)


Reviewer 4

Summary

This paper synthesizes recent ideas from online learning and optimization into a new algorithm for bandit convex optimization that efficiently obtains O~(T^(11/16)) regret for lipschitz functions and O~(T^(8/13)) regret for functions with lipschitz gradients. This improves on the previous T^(3/4) regret for lipschitz functions and T^5/8 regret for functions lipschitz gradients. To achieve this, the authors utilize the predictable sequence / optimistic learning framework of Rakhlin and Sridharan (2013), combined with the smoothed gradient estimate technique of Dekel et al. (2015), as well as self-concordant regularization.

Qualitative Assessment

This paper is clearly written and technically sound. It was nice to see the way these techniques fit together to produce an improved algorithm. The way the references were handled makes it difficult to evaluate the significance of this approach. In particular, the authors seem to dismiss the recent line of work starting Bubeck/Eldan 2015 and Bubeck et al. 2015 that nonconstructively establishes the existence of algorithms obtaining \sqrt{T} regret for BCO -- this work is mentioned in the introduction but never discussed in detail. The line "However, this paper is unpublished and to the best of our knowledge still unverified by the community" -- in reference to the recent algorithm of Hazan and Li that obtains \sqrt{T} regret with suboptimal dependence on dimension in both running time and regret --strikes me as unprofessional. It would have been much more satisfying to see a qualitative comparison with this line of research to see what can be gained by combining its ideas with the ideas from this paper. Overall, the results of the paper at hand seem incremental compared to those from the Bubeck line of research -- I feel that the recent manuscript by Bubeck/Eldan/Lee efficiently obtaining \sqrt{T} regret with polynomial dependence on dimension confirms this view.

Confidence in this Review

2-Confident (read it all; understood it all reasonably well)


Reviewer 5

Summary

In this paper, the authors try to improve the regret bounds of bandit convex optimization for both Lipschitz continuous functions and functions with Lipschitz continuous gradients. The key idea, as nicely articulated in Section 4.1, is to come up with a different estimate of the gradient at each round of iterations. With this small change, the author are able to introduce a factor of O(1/k^2), instead of O(1/k), in the bound, which accounts for the final improvement.

Qualitative Assessment

The main content of this work combines the idea from Dekel 2015 and the idea from Rakhlin 2013. Overall the paper is well written and is easy to follow. The improvement in the regret bound of bandit optimization seems to be significant, although the analysis seems to be a relatively straightforward combination of Lemma 4 and the analysis fro Dekel 2015, which leads to a relatively low rating on novelty and impact.

Confidence in this Review

2-Confident (read it all; understood it all reasonably well)


Reviewer 6

Summary

This paper presents a new algorithm for the bandit convex optimization problem improving the state of the art regret bound to O(T^{8/13} d^{5/3}). Theoretical analysis are also presented. The algorithm works by using a self concordant barrier as regularization function together with ideas from optimistic follow the regularized leader.

Qualitative Assessment

1- Please define int(..) before using it a line 60 page 2. 2- Does Theorem 3 means that y_t defined in Figure 1 line 3 is always inside the bounded domain? If yes, you should explain it clearly. Otherwise how to make sure y_t is always inside the domain? 3- How does one construct such a set concordant barrier for a given problem instance? This is very important for OPTIMISTICBCO and a technique or algorithm that can computes such barrier function for any problem instance should be provided by the paper. Without that the algorithm is not really useful and won't be usable in practice.

Confidence in this Review

2-Confident (read it all; understood it all reasonably well)